# Nitrogen Oxide Removal by Coal-Based Activated Carbon for a Marine Diesel Engine

**Zongyu Wang [1], Hailang Kuang [1], Jifeng Zhang [1,2], Lilin Chu [1] and Yulong Ji [1,*]**

[1] College of Marine Engineering, Dalian Maritime University, Dalian 116026, China;
wangzongyu09@163.com (Z.W.); kuanghailang_dlmu@163.com (H.K.); zhangjifeng@dlmu.edu.cn (J.Z.);
chulilinlove@outlook.com (L.C.)

[2] Yangtze Delta Region Institute of Tsinghua University, Zhejiang, Jiaxing 314006, China

[*] Correspondence: jiyulong@dlmu.edu.cn; Tel.: + 86-411-84724306

**Abstract:** Vanadium-based catalysts are mainly used for marine diesel exhaust denitration. However, their poor catalytic ability at low temperature and poor sulfur tolerance, as well as high toxicity and cost, are big turnoffs. AC (Activated carbon) exhibits good adsorption capacity and catalytic ability in denitration because of its high specific surface area and chemical activity. In this paper, coal-based AC was used for simulating diesel exhaust denitration in different conditions. The results show that the NO removal ability of AC is poor in an $NO/N_2$ system. The $NO_2$ removal ability is excellent in an $NO_2/N_2$ system, where NO is desorbed. The NO$x$ removal efficiency is 95% when the temperature is higher than 200 °C in an $NO_2/NH_3/N_2$ system. When the temperature is lower than 100 °C, AC can catalytically oxidize NO to $NO_2$ in an $NO_2/O_2/N_2$ system. The near-stable catalytic efficiencies of AC for a slow SCR (Selective Catalytic Reduction) reaction, a standard SCR reaction, and a fast SCR reaction at 300 °C are 12.1%, 31.6%, and 70.8%, respectively. When ships use a high-sulfur fuel, AC can be used after wet scrubber desulfurization to catalytically oxidize NO to $NO_2$ at a low temperature. When ships use a low-sulfur fuel, AC can be used as a denitration catalyst at high temperatures.

**Keywords:** coal-based activated carbon; marine diesel engine; exhaust denitration; SCR reaction; oxygen-containing functional group

## 1. Introduction

At present, diesel engines are widely used in ships as the main power unit and generation unit. Marine diesel engines generally use high-sulfur, heavy fuel oil. In addition to $CO_2$ contributing to the greenhouse effect, the exhaust gas also contains lots of SOx, NOx, particulate matter (PM), and other pollutants. The International Maritime Organization (IMO) has issued MARPOL 73/78 Annex VI [1] to control the emissions of SOx and NOx from marine diesel engine exhaust. The current means to meet the Convention's SOx emission requirements mainly include low-sulfur fuel oil and wet scrubber desulfurization technology. The alkaline liquid used in a wet scrubber includes NaOH [2], $Mg(OH)_2$ [3,4], seawater [5,6], and so on. The means to meet the Convention's Tier III NOx emission requirements are mainly exhaust gas recirculation (EGR) [7,8] and selective catalytic reduction (SCR) [9]. EGR technology is generally only adopted by diesel engine manufacturers because of its high technical barriers and changes in the diesel engine body. SCR is currently the most widely used exhaust gas denitration technology, and often uses vanadium-based catalysts. The operating temperature of vanadium-based catalysts is 250–400 °C, and its denitration efficiency is generally up to 90% [10–12]. From the above, there are three possible technical routes that can finally achieve the desired SOx and NOx removal:

(1) Low-sulfur fuel oil → vanadium-based SCR denitration. This route is simple and effective. However, the fuel cost is expected to rise because of the high price of low-sulfur fuel oil. The fuel supply system and lubrication system of the diesel engine also need to be modified. In addition, the vanadium-based catalyst is expensive and highly toxic. It causes secondary pollution at disposal after the catalyst's life has expired.

(2) High-sulfur fuel oil → vanadium-based SCR denitration → wet scrubber desulfurization. The fuel cost is low by this technical route, but the SOx in the exhaust gas may corrode the subsequent SCR denitration reactor. Secondly, the use of high-sulfur fuel may lead to an increase in PM emissions, and the PM often adheres to the catalyst surface, causing a decrease in denitration efficiency and an increase in exhaust back pressure. What is more, sulfur poisoning of the vanadium-based catalyst is a known issue [13].

(3) High-sulfur fuel oil → wet scrubber desulfurization → vanadium-based SCR denitration. This route can avoid the poisoning effect of SOx on the denitration catalyst. However, the temperature of the exhaust gas after wet scrubber desulfurization is generally below 100 °C, even as low as 50 °C [14]. The denitration efficiency of vanadium-based catalyst at low temperatures is poor. It needs a heat exchanger to reheat the exhaust gas before the exhaust gas enters the SCR reactor, which makes the system more complicated.

The fuel price is the most important factor in the choice of desulfurization technology. However, SCR technology seems indispensable for ship denitration. There are mainly three reactions in SCR denitration, shown as Equations (1)–(3) [11] below. Equation (1) has the slowest reaction rate, also known as the slow SCR reaction. NOx in diesel exhaust is mainly NO and $NO_2$, of which NO accounts for more than 90%. The reaction activation energy of Equation (2) is lower than that of Equation (3), so the reaction rate of Equation (2) is higher. Equation (2) is also called the fast SCR reaction. $O_2$ is much more prevalent than $NO_2$ in exhaust gas. So Equation (3) is dominant in the SCR reaction, also known as the standard SCR reaction. [10]. When $NO/NO_2$ is equal to 1, the reaction rate is the fastest. Increasing the $NO_2/NOx$ ratio appropriately helps to increase the SCR reaction rate [15].

$$4NH_3 + 6NO \rightarrow 5N_2 + 6H_2O \tag{1}$$

$$2NH_3 + NO + NO_2 \rightarrow 2N_2 + 3H_2O \tag{2}$$

$$4NH_3 + 4NO + O_2 \rightarrow 4N_2 + 6H_2O \tag{3}$$

In summary, it is important to look for a denitration catalyst that is inexpensive, has high denitration efficiency in a wide temperature range and causes no secondary pollution. Activated carbon is not only cheap and has a variety of sources, but also has good adsorption capacity and catalytic ability due to the high specific surface area and chemical activity. Some metal oxides can be impregnated onto the AC surface to further increase the denitration efficiency at high temperatures [16]; therefore, it has great application prospects in the future. There are many factors affecting the denitration efficiency of AC, including the AC type, specific surface area (mainly related to physical adsorption) and surface functional groups (mainly related to chemical adsorption and catalysis), NOx concentration, $O_2$ concentration, exhaust gas temperature, adding $NH_3$ or not, and so on. At present, there are three main types of commercial ACs, i.e., coal-based, coconut shell, and wood-based. The wood-based AC is mostly used for water treatment, and is not considered environmentally friendly because it needs to consume wood. The price of coconut shell AC is roughly twice that of coal-based AC [17]. Therefore, this paper chooses coal-based AC as the research object. Since the marine diesel engines could use different sulfur content fuels, the desulfurization means are also different, which leads to different temperatures of the exhaust gas during denitration. This paper mainly studies the performance of AC to catalyze the oxidation of NO to $NO_2$ at low temperatures and as the SCR catalyst at high temperatures, and explores the application prospects of AC in the field of ship denitration.

## 2. Experimental System and Methods

### 2.1. Experimental System

In order to study the effects of different gas compositions, concentrations, and temperatures on the denitration ability of the coal-based AC, a simulating gas supply system is used to simulate real diesel exhaust gas in this paper. The experimental system is shown in Figure 1. It mainly includes a gas supply unit, quartz reactor, temperature control unit, flue gas analyzer (Testo 350, Freiburg, Germany), and exhaust gas absorption device. The gas supply unit mainly includes gas cylinders, pressure reducing valves, mass flow controllers (Seven Star CS200AD, Beijing, China), mixing chamber, etc. The temperature control unit includes a tubular heating furnace (Ruipu Electric Equipment Factory, 220V/3kW, Yangzhou, China) and temperature controller. $N_2$ and $O_2$ are high-purity standard gases. NO is a 10% standard gas with $N_2$ as the carrier gas, and so are $NO_2$ and $NH_3$. The concentrations of each gas component in the mixed gas are controlled by the mass flow controllers. The quartz reactor, a hollow cylinder of 26 mm inner diameter, 2 mm wall thickness, and 300 mm length, is placed in the tubular heating furnace. The experimental temperature is controlled by a temperature controller with $\pm 1\,^\circ C$ accuracy. The flue gas analyzer can continuously monitor the temperature and the concentrations of NO, $NO_2$, and CO of the mixed gas with a sampling period of 1 s by an electrochemical method. In order to prevent the exhaust gas from polluting the air, the gas passing through the flue gas analyzer is sent to an exhaust gas absorption device for treatment.

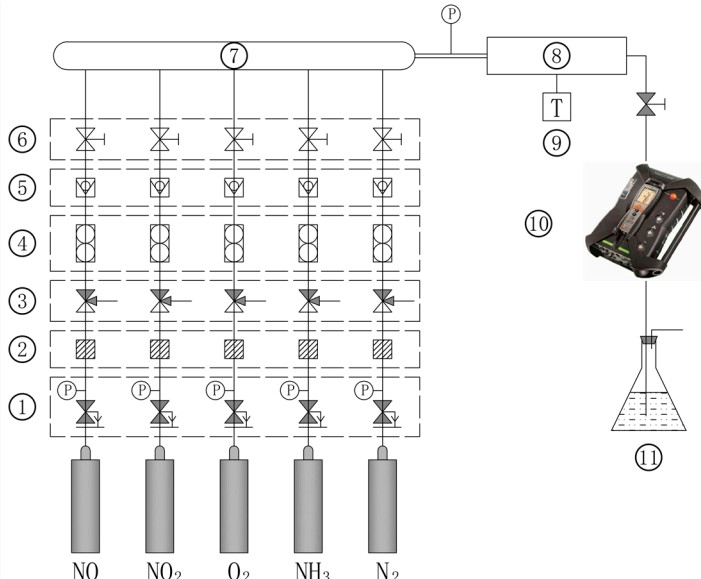

**Figure 1.** Experimental system of activated carbon denitration. 1. Pressure reducing valve 2. Filter 3. Three-way valve 4. Mass flow controller 5. Check valve 6. Stop valve 7. Mixing chamber 8. Tubular heating furnace 9. Temperature controller 10. Flue gas analyzer 11. Exhaust gas absorption device.

### 2.2. Materials and Methods

The coal-based columnar AC used in this paper was purchased from Henan Huanyu Carbon Co., Ltd., Xuchang, China. Its diameter, length, and specific surface area are 1.5 mm, 2–4 mm, and 326 $m^2$/g, respectively. The AC was washed with deionized water before the experiments to remove impurities on the surface and then dried in a drying oven at 110 $^\circ C$ for 12 h. The types and contents of oxygen-containing functional groups on the AC surface were detected by the Boehm titration method. The total gas volume was 2 L/min with $N_2$ as the carrier gas and 30 g of AC was used in each group of experiments. (The corresponding volume of gas per volume of catalyst and per hour, i.e., VVH, was 2500 $h^{-1}$.) The exhaust gas temperature of large low-speed two-stroke marine diesel engines is

generally lower than 300 °C [4]. Therefore, the temperature range studied in this paper was from room temperature (15 °C) to 300 °C.

The NOx concentration in the exhaust gas of marine diesel engine is generally about 1000 μL/L, and NO accounts for the main part. When the NO/NO$_2$ molar ratio is 1, the fast SCR reaction Equation (2) takes place, and the reaction rate is the fastest. So, the concentrations of NO and NO$_2$ were both 500 μL/L in the fast SCR reaction. The total concentration of NO and NO$_2$ remained 1000 μL/L. When NH$_3$ was needed, NH$_3$ was added at an ammonia-nitrogen ratio of 1 depending on the initial NOx. The O$_2$ concentration was 5% when needed. We do not pay special attention to the physical adsorption time of AC in this paper. The discussion of the denitration efficiency of AC in the following refers to the removal efficiency when the outlet NOx concentration is in a near-stable condition. The near-stable denitration efficiency is then defined as follows:

$$\eta = \frac{C_{in} - C_{out}}{C_{in}} \times 100\% \tag{4}$$

Here $\eta$ is the NOx removal efficiency of AC, %; $C_{in}$ is the inlet NOx concentration, μL/L; $C_{out}$ is the outlet NOx concentration, μL/L.

## 3. Results and Discussion

### 3.1. Effect of AC on Single-Component NO

The NO adsorption on the surface of AC is the premise of AC denitration. The temperature has a great influence on the AC denitration ability. Therefore, this paper first studied the removal ability of AC on single component NO in the range of 15–300 °C. As shown in Figure 2, there are four stages according to the temperature, which are 15 °C, 15–100 °C, 100–200 °C, and 200–300 °C, and the inlet gas is 1000 μL/L NO/N$_2$. The outlet NO reaches 970 μL/L within 10 min at 15 °C. Modified activated carbon fiber (ACF) was used to investigate ACF's denitration ability at 40 °C. When the inlet gas is 900 μL/L NO/N$_2$, the outlet NO reaches 800 μL/L within 5 min and then increases slowly [17]. Our results are similar to his. We know that low temperature contributes to the physical adsorption of AC, and generally ACF has a larger specific surface area than AC. So ACF's physical adsorption ability is stronger than AC's. However, the NO adsorption amounts of both ACF and AC are still limited at low temperatures. With the increase in temperature, the physical adsorption ability of AC gradually deteriorates, and the chemisorption ability and catalytic ability gradually improve. Therefore, in order to improve the denitration ability of AC, it is also necessary to optimize its catalytic ability.

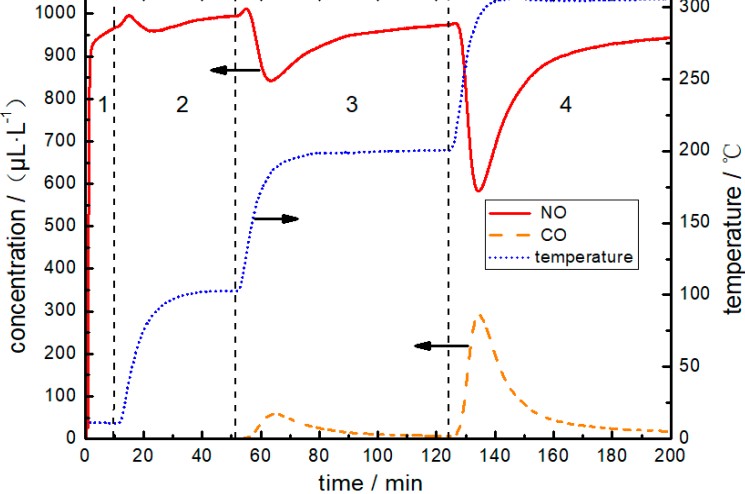

**Figure 2.** Effect of activated carbon in NO/N$_2$ system (1000 μL/L NO).

During the heating process, the outlet NO will increase slightly due to the partial desorption of the adsorbed NO. However, as the temperature increases, the chemisorption ability of AC increases, and more NO is absorbed by chemisorption. Therefore, the outlet NO is decreased. When the temperature is kept constant, the adsorption-activated sites on the surface of the AC gradually decrease over time, leading to an increasing concentration of NO at the outlet. However, as the temperature gradually increases from 15 °C to 300 °C, the catalytic ability of AC increases, and so does the denitration efficiency. It can be seen from Figure 2 that the near-stable denitration efficiencies of AC at 100 °C, 200 °C, and 300 °C are 2%, 3.6%, and 5.8%, respectively.

The chemisorption and catalytic abilities of AC are closely related to the oxygen-containing functional groups on the surface of AC [18]. The contents of oxygen-containing functional groups on the surface of AC were measured by the Boehm titration method [19–21] in this paper, and the results are shown in Table 1. It can be seen that the coal-based AC is acidic, and the phenolic hydroxyl group has a relatively higher proportion. Several papers [22–24] studied the decomposition temperature of different oxygen-containing functional groups on the surface of AC. It can be seen from Figure 2 that CO is generated when the temperature is between 100 °C and 200 °C. However, when the temperature is kept at 200 °C, the CO concentration basically does not change. In the temperature range of 100–200 °C, it is mainly the carboxyl group decomposing [23]. When the temperature is between 200 °C and 300 °C, the experimental phenomenon is similar to that of the third stage, and both the CO concentration and the denitration efficiency are higher. In the temperature range of 200–300 °C, this is mainly due to the decomposition of the carboxyl group and the lactone group [23,24]. The decomposition temperature of the phenolic hydroxyl group is generally higher than 600 °C [24], so it does not decompose under the experimental conditions herein. These oxygen-containing functional groups may react with NO, such as Equation (5) generating CO and Equation (6) generating $CO_2$ ($C_f$ represents the activated sites). CO also has a certain degree of reducibility, and will react with NO, i.e., Equation (7), which also lowers the concentration of NO. In general, the denitration efficiency of AC increases slightly with the temperature when the inlet is $NO/N_2$. However, there is still a big gap toward the actual applications of AC for exhaust gas denitration. Therefore, other methods should be considered.

$$2C_f + 2NO \rightarrow N_2 + 2CO \tag{5}$$

$$C_f + 2NO \rightarrow N_2 + CO_2 \tag{6}$$

$$2CO + 2NO \rightarrow N_2 + 2CO_2 \tag{7}$$

**Table 1.** Surface chemical characteristics of activated carbon (Unit: mmol/g).

| Species | Carboxyl | Lactone | Phenolic Hydroxyl | Acidic Groups | Basic Groups | Total Contents |
|---------|----------|---------|-------------------|---------------|--------------|----------------|
| Value | 0.150 | 0.045 | 0.233 | 0.428 | 0.278 | 0.705 |

### 3.2. Effect of AC on NO + NH₃

When the temperature is lower than 300 °C, the denitration ability of AC as the reductant was poor. To further improve the denitration ability of AC, $NH_3$ is generally added as the reductant [25,26]. As shown in Figure 3, the denitration ability of AC at different temperatures is investigated in this paper when the inlet gas is 1000 μL/L NO + 1000 μL/L $NH_3$, i.e., slow SCR reaction Equation (1). There are two explanations for the catalytic denitration mechanism of AC when $NH_3$ is present. One is the Eley-Rideal mechanism (E-R), in which the adsorbed $NH_3$ reacts with the gas phase NO [27], and the other is the Langmuir-Hinshelwood mechanism (L-H), in which the adsorbed $NH_3$ reacts with adsorbed NO [28]. Both mechanisms suggest that the $NH_3$ adsorption process on the surface of AC is the rate-limiting step of Equation (1). Therefore, more acidic oxygen-containing functional groups on the surface of AC can contribute to the adsorption of $NH_3$, which increases the reaction rate of

denitration. Figure 4, which is supported by [28], shows the L-H mechanism of NO and $NH_3$ reacting on the surface of AC, and $NH_3$ is mainly adsorbed on the phenolic hydroxyl group. The coal-based AC used in this paper is acidic and the phenolic hydroxyl-containing functional group is relatively high, which promotes denitration. It can be seen from the comparison between Figures 2 and 3 that the trend and concentration of CO after adding $NH_3$ are basically the same as without $NH_3$, indicating that $NH_3$ has no effect on the decomposition of the oxygen-containing functional groups. After adding $NH_3$, NO can be converted to $N_2$ by slow SCR reaction Equation (1), and the denitration rate of AC is slightly increased. The near-stable denitration efficiencies of AC at 15 °C, 100 °C, 200 °C, and 300 °C are 10.5%, 9.1%, 10%, and 12%, respectively, which are slightly higher than without $NH_3$. However, the catalytic effect of AC on the Equation (1) is still small.

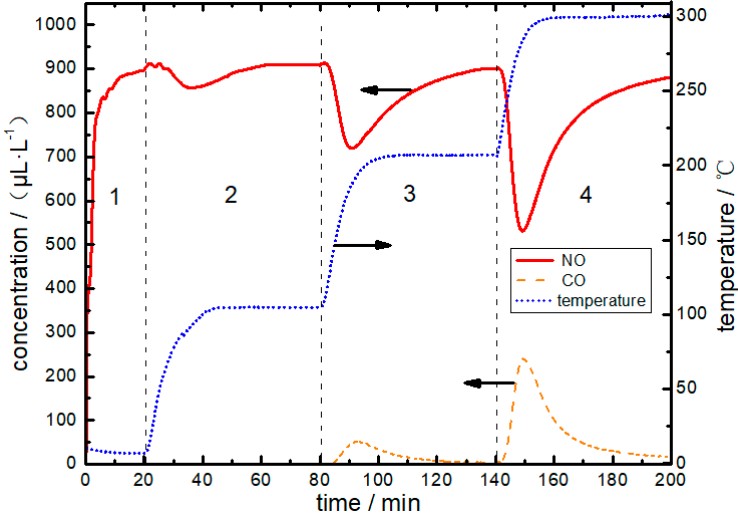

**Figure 3.** Effect of AC in $NO/NH_3/N_2$ system (1000 μL/L NO + 1000 μL/L $NH_3$).

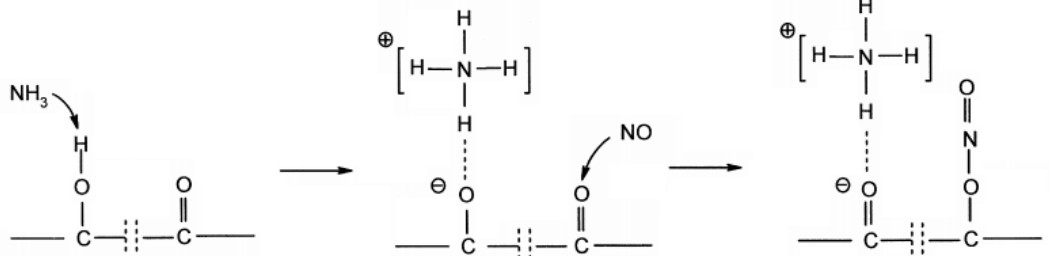

**Figure 4.** Reaction mechanism of NO reduction with $NH_3$ on the AC surface.

### 3.3. Effect of AC on Single-Component $NO_2$ and $NO_2$ + $NH_3$

The exhaust gas of diesel engine contains a certain concentration of $NO_2$. The effects of AC on single-component $NO_2$ and $NO_2$ + $NH_3$ are the basis for the study of fast SCR reaction Equation (2). Therefore, this section will discuss the effects of AC on single $NO_2$ and $NO_2$ + $NH_3$ separately.

As shown in Figure 5, the experiment is divided into four stages, namely, 15 °C (1st stage), 15–100 °C (2nd stage), 100–200 °C (3rd stage), and 200–300 °C (4th stage). The inlet gas is 500 μL/L $NO_2/N_2$. $NO_2$ is not detected in the outlet during the experiment, which indicates that the $NO_2$ removal ability of AC is excellent at 15–300 °C. Interestingly, NO is detected in the outlet and the concentration of NO increases gradually with the temperature. There is a large amount of NO desorbed during the heating process. During the process of heating from 100 °C to 200 °C, the concentration of desorbed NO can reach up to 2800 μL/L. When the temperature remains stable at 200 °C and 300 °C, respectively, $NO_2$ is almost completely converted to NO, which is then desorbed. CO and $CO_2$ may be produced

when the temperature is above 100 °C. Since the flue gas analyzer cannot measure $CO_2$, the amount of $CO_2$ needs further study. The likely reactions are Equations (8)–(10) [29,30].

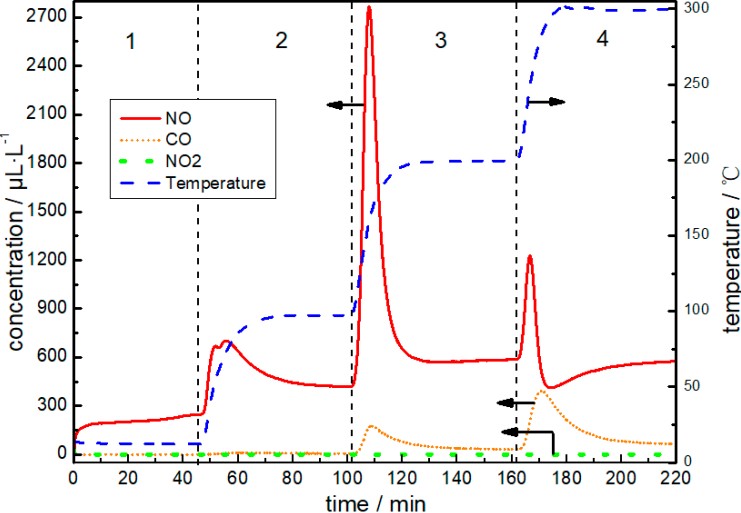

**Figure 5.** Effect of AC on $NO_2$.

Then we carried out the experiment with $NO_2 + NH_3$. As shown in Figure 6, the experiment is divided into five stages, namely, 15 °C without AC (1st stage), 15 °C with AC (2nd stage), 15–100 °C with AC (3rd stage), 100–200 °C with AC (4th stage), and 200–300 °C with AC (5th stage). The inlet gas is 500 μL/L $NO_2$ + 500 μL/L $NH_3/N_2$. $NH_3$ can react with $NO_2$ at room temperature [31]. It can be seen from Figure 6 that $NO_2$ gradually decreases without AC in the first stage, that is, the reaction takes place between $NO_2$ and $NH_3$, and the products include $N_2$ and NO. When the gas is passed into the AC reactor (2nd-5th stages), $NO_2$ almost completely disappears. During the heating process, NO desorbs, resulting in an increase in the concentration of NO at the outlet. When the temperature is stable, the temperature is higher and the lower the outlet NO is acquired. When the temperature is higher than 200 °C, the removal efficiency of NOx by AC is more than 95% in the presence of $NH_3$.

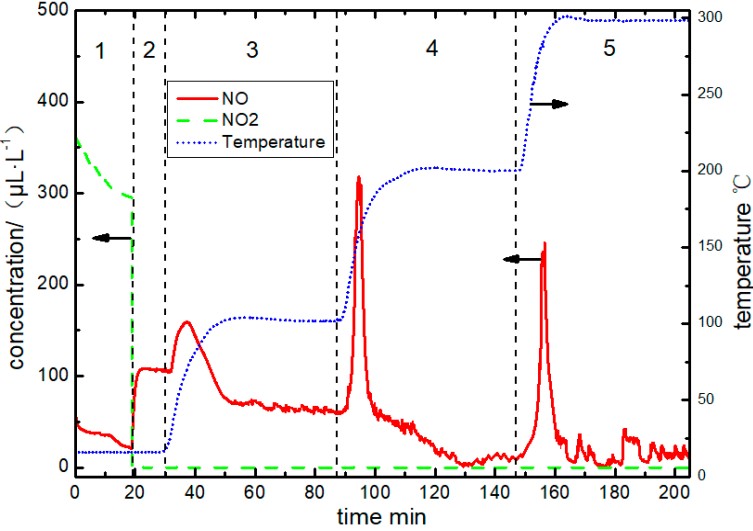

**Figure 6.** Effect of AC on $NO_2 + NH_3$.

In summary, this study suggests that when there is no $NH_3$, AC converts $NO_2$ into NO, and the concentration of NO in the outlet increases with temperature. When $NH_3$ is added, AC catalyzes $NO_2$ and $NH_3$ to generate $N_2$, i.e., Equation (11). The catalytic activity of AC increases gradually with the

temperature. In addition, the desorbed NO participates in the rapid SCR reaction Equation (2), thereby further improving the denitration ability of AC.

$$C_f + NO_2 \rightarrow NO + C_f\text{-}O \tag{8}$$

$$C_f + 2NO_2 \rightarrow 2NO + CO_2 \tag{9}$$

$$Cf + NO_2 \rightarrow NO + CO \tag{10}$$

$$8NH_3 + 6NO_2 \rightarrow 7N_2 + 12H_2O \tag{11}$$

### 3.4. Effect of AC on NO + NO₂ + NH₃

When the $NO/NO_2$ molar ratio is 1, SCR denitration mainly takes place according to the fast SCR reaction Equation (2). At present, the catalytic effect of AC on Equation (2) at different temperatures is not clear. Therefore, this section mainly studies the catalysis of AC for Equation (2) at 15–300 °C.

The inlet gas is 500 μL/L NO + 500 μL/L $NO_2$ + 1000 μL/L $NH_3$. As shown in Figure 7, the experiment is divided into six stages, namely, 15 °C without AC (1st stage), 15 °C with AC (2nd stage), 15–100 °C with AC (3rd stage), 100–200 °C with AC (4th stage), 200–300 °C with AC (5th stage), and 300 °C with AC and without $NH_3$ (6th stage). It can be seen from Figure 7 that $NO_2$ can react with $NH_3$ to generate NO and $N_2$ at room temperature without AC in the 1st stage. In the 2nd stage, there is almost no $NO_2$ at the outlet when the gas is introduced into the activated carbon reactor. The concentration of the outlet NO is also reduced to some extent. The denitration efficiency is about 59.5% at 15 °C. In the 3rd stage, when the temperature is raised from 15 °C to 100 °C, the chemical denitration of AC is not significantly improved, but its physical adsorption ability is greatly reduced, resulting in an increase in outlet NO, and the denitration efficiency is about 53%. In the 4th stage, when the temperature is raised from 100 °C to 200 °C, the chemical denitration of AC is greatly enhanced, around 69.2%, and the outlet NO is greatly reduced. In the 5th stage, when the temperature is further increased from 200 °C to 300 °C, the denitration ability of AC does not change significantly, and the near-stable denitration efficiency reaches 70.8%. In the 6th stage, when $NH_3$ is turned off, most $NO_2$ is converted to NO and then desorbed, resulting in a rapid NO concentration increase at the outlet, eventually reaching 920 μL/L. This indicates that $NH_3$ plays an important role in the denitration ability of AC at high temperatures. It is found that $NH_3$ increases the denitration efficiency of AC from 8% to 70.8% at 300 °C.

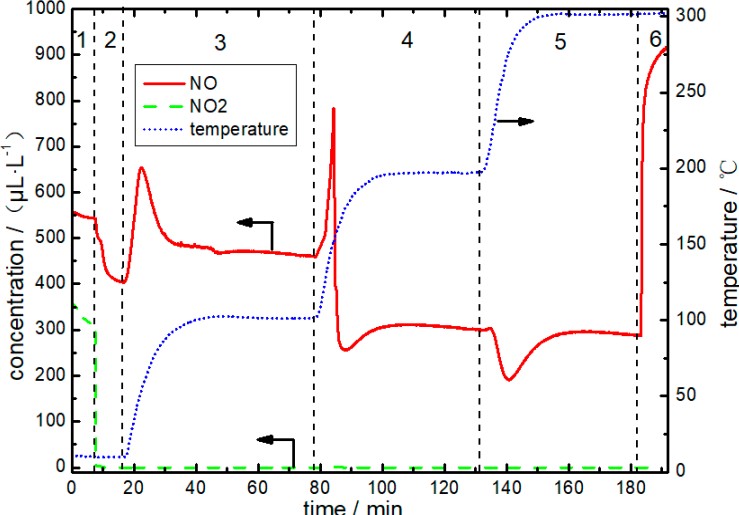

**Figure 7.** Effect of AC on NO + NO₂ + NH₃.

### 3.5. Effect of AC on NO + O₂ and NO + O₂ + NH₃

The diesel engine exhaust gas contains a large amount of $O_2$. $O_2$ can not only oxidize NO to $NO_2$, but also is the reactant of standard SCR reaction Equation (3). Therefore, $O_2$ will have a great influence on the denitration ability of AC. When the inlet gas is 1000 μL/L NO + 5% $O_2$, the denitration ability of AC at different temperatures is shown in Figure 8, which shows that it takes 660 min for outlet NOx to reach 900 μL/L at 15 °C and 500 min at 50 °C. Compared with the experiment without $O_2$ at 15 °C (Figure 2), the adsorption saturation time of AC is greatly prolonged, indicating that $O_2$ can greatly promote the chemical adsorption of NO on the surface of AC. In addition, $NO_2$ starts to appear at the outlet after 300 min at 15 °C. Then $NO_2$ gradually increases and finally stabilizes, while NO basically maintains about 300 μL/L. When the temperature is 50 °C, $NO_2$ starts to appear at the outlet after 200 min and since then NO maintains about 480 μL/L. The trends of NO and $NO_2$ are similar to that at 15 °C. The above experimental phenomenon is mainly due to the different adsorption sites ($C_f$-A and $C_f$-B) on the surface of AC for NO and $NO_2$. $C_f$-A can adsorb NO and $NO_2$, but it is easier to adsorb NO, while $C_f$-B can adsorb $NO_2$ only [27]. At the beginning of the experiment, gaseous NO can be directly adsorbed on the oxygen-containing functional groups and then converted into $C_f$-A-$NO_2$, or it can be adsorbed on the activated sites to form $C_f$-A-NO, and then oxidized to $C_f$-A-$NO_2$ by $O_2$. Due to the combination force of $NO_2$ on $C_f$-A-$NO_2$ being relatively small, $NO_2$ is easy to desorb or directly transfer to the adjacent $C_f$-B to form $C_f$-B-$NO_2$. When $C_f$-B is saturated, the desorbed $NO_2$ will be out of the reactor. Finally, the adsorption and oxidation of NO balance the adsorption and desorption of $NO_2$ dynamically. The outlet concentrations of NO and NO2 remain roughly stable. As can be seen from Figure 8, the conversion rates of AC to catalyze the oxidation of NO to $NO_2$ at 15 °C and 50 °C are 62.3% and 43.8%, respectively.

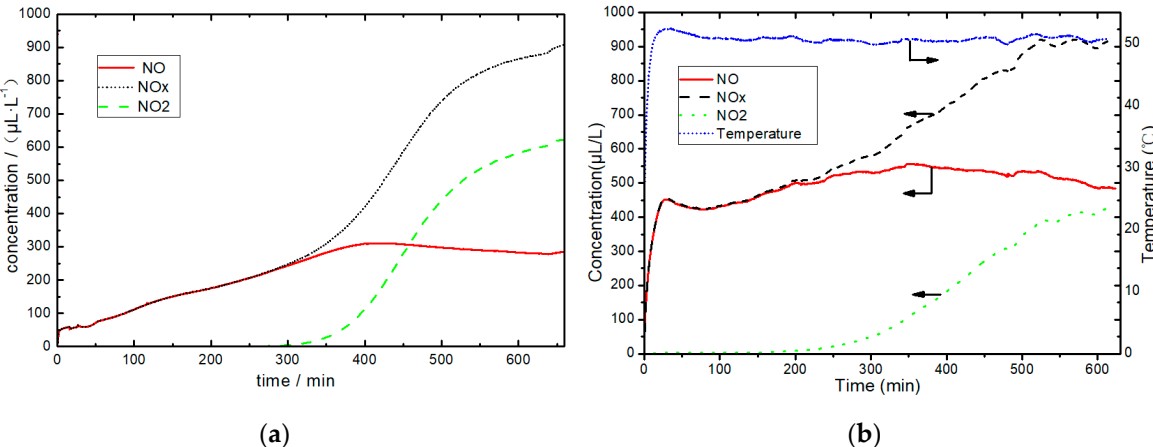

(**a**)                                          (**b**)

**Figure 8.** Effect of AC on NO + $O_2$ at low temperatures: (**a**) 15 °C; (**b**) 50 °C.

When ships use low-sulfur fuel oil, the wet scrubber desulfurization will no longer be needed and the temperature of the exhaust gas entering the SCR reactor is higher than that coming out of the scrubber in a high-sulfur case. Therefore, it is necessary to study the denitration ability of AC at higher temperatures. As shown in Figure 9, the denitration ability of AC are simulated with or without $NH_3$ at 100 °C, 200 °C, and 300 °C, respectively. Each experiment is divided into three stages: the first stage is the heating process (from room temperature to the set temperature), and the second and third stages are maintained at the set temperature. In the first stage and the third stage, the inlet gas is 1000 μL/L NO + 5% $O_2$, and in the second stage, 1000 μL/L $NH_3$ is added. It can be seen from Figure 9 that in the first 60 min, the NO concentration in the outlet rises rapidly during the heating process. The faster the heating rate, the higher the NO concentration in the outlet. When the temperature is stable, the outlet NO gradually decreases and eventually stabilizes. Compared with Figure 8, the adsorption saturated time of AC is shorter when the temperature is higher than 100 °C. After the adsorption

is saturated, NO will not be oxidized to $NO_2$, indicating that the catalytic oxidation of NO by AC is poor when the temperature is higher than 100 °C. At this time, the stable denitration efficiencies of AC at three temperatures are 12.6%, −0.5%, and 2.5%, respectively. Illán-Gómez [32] carried out temperature-programmed reduction (TPR) experiments using coal-based activated carbon with 0.5% NO + 5% $O_2$/He from room temperature to 500 °C. The results show that the denitration rate of AC decreases and then increases with the temperature. The denitration rate is basically negative between 150 and 300 °C, about −20% at 220 °C. Shu [33] used sawdust, rice husk, and corncob-activated carbon to carry out TPR experiments with 0.1% NO + 5% $O_2$/$N_2$ from room temperature to 600 °C. The results show that the trend of denitration efficiency is basically the same as that of Illán-Gómez, and the denitration efficiency is negative at 150–350 °C. The experimental results obtained in this paper are basically the same as those Illán-Gómez and Shu obtained via the TPR method. The experimental phenomenon is mainly caused by the dynamic balance between the better adsorption ability of AC at low temperature and the better reduction and catalytic ability at high temperature. When the temperature is lower than 200 °C, the adsorption ability is dominant. When the temperature is higher than 200 °C, the reduction and catalytic ability are dominant. However, since the diesel engine exhaust gas temperature is generally lower than 300 °C, the denitration ability of AC is not ideal when no reductant is added. At the 60th minute, $NH_3$ is added to simulate standard SCR reaction Equation (3). The NO concentrations of the outlet at three temperatures decrease rapidly. When the $NH_3$ is stopped, the NO concentration of the outlet increases rapidly. When the temperatures are 100 °C, 200 °C, and 300 °C, the corresponding near-stable denitration efficiencies are 34.6%, 16%, and 31.6%, respectively. Experimental results show that $NH_3$ can greatly improve the denitration ability of AC in the range of 100–300 °C.

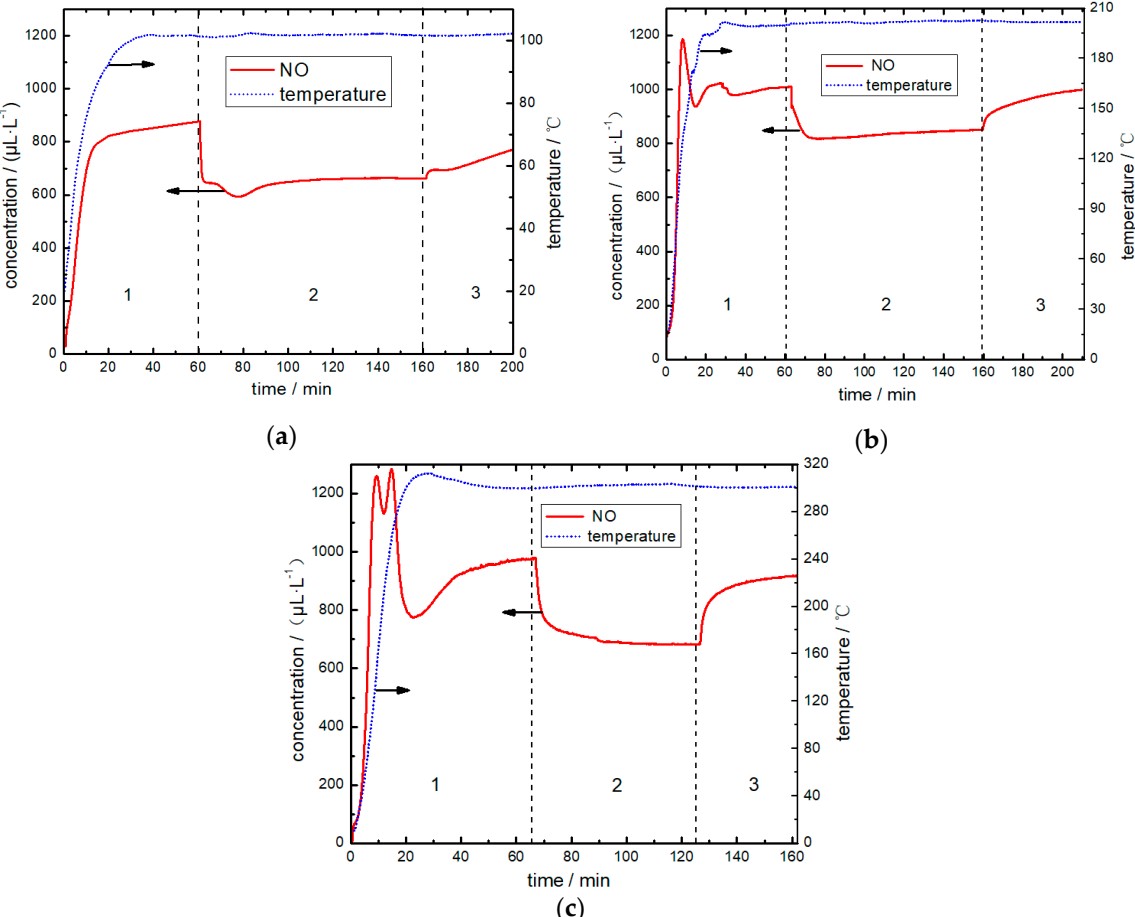

**Figure 9.** Effect of AC on NO + $O_2$ + $NH_3$ at high temperature. (**a**) 100 °C; (**b**) 200 °C; (**c**) 300 °C.

*3.6. The Application Prospect of AC in Ship Exhaust Gas Denitration*

When ships use low-sulfur fuel oil, the temperature of the exhaust gas entering the SCR denitration reactor is generally 200-300 °C. From the above experimental results, we know that the near-stable denitration efficiencies of the slow SCR reaction Equation (1), the fast SCR reaction Equation (2), and the standard SCR reaction Equation (3) under the action of activated carbon are 12.1%, 70.8%, and 31.6%, respectively. Increasing the proportion of $NO_2$ in the exhaust gas can increase the denitration efficiency of AC. In addition, the AC may be modified or loaded with other metal oxides to further increase the denitration efficiency. When ships use high-sulfur fuel, we know from the above experimental results that the exhaust gas cooled to about 50 °C after wet scrubber desulfurization can be passed into the activated carbon oxidizer to oxidize the NO to $NO_2$. As $NO_2$ is easily soluble in water, the gas can be introduced into the washing tower again for denitration treatment. Therefore, it does not need the heat exchanger to reheat the exhaust gas. What is more, ammonia- and vanadium-based catalysts are no longer needed. The activated carbon is cheaper and has a wide range of sources. In the future, based on changes in oil prices, whether using low-sulfur fuel oil or high-sulfur fuel oil with wet scrubber to meet ships' sulfur emission requirements, SCR will be an effective method of ship denitration. Activated carbon can be used as an oxidation catalyst in low-temperature exhaust gas and as an SCR denitration catalyst in high-temperature exhaust gas. It will have good application prospects in the field of ship exhaust gas denitration in the future.

## 4. Conclusions

Based on the above experiments on the denitration ability of coal-based activated carbon under different conditions, the paper mainly draws the following conclusions:

(1) In the range of 15-300 °C, when the inlet gas is $NO/N_2$, the removal of NO by coal-based activated carbon mainly depends on the adsorption, and the near-stable denitration efficiency is low. When the inlet gas is $NO_2/N_2$, $NO_2$ is finally converted to NO and desorbs from the surface of activated carbon. However, when $NH_3$ is added and the temperature is higher than 200 °C, the NOx removal efficiency is up to 95% in the $NO_2/N_2$ system.

(2) When the inlet gas is 1000 µL/L NO + 5% $O_2$, the saturated adsorption time of NO for AC at 10 °C and 50 °C is 660 min and 500 min, respectively. However, the adsorption saturation time drops sharply with an increase in temperature. When the temperature is above 200 °C, the adsorption saturation time is less than 10 min. When the temperature is lower than 100 °C, activated carbon can catalyze the oxidation of NO to $NO_2$. The conversion efficiency of NO to $NO_2$ at 15 °C and 50 °C is 62.3% and 43.8%, respectively. When the temperature is higher than 100 °C, the ability of AC for catalytic oxidation of NO is minimal.

(3) The stable denitration efficiencies of activated carbon with different inlet gas are investigated at 100 °C, 200 °C, and 300 °C. When the inlet gas is $NO/NH_3/N_2$, simulating slow SCR reaction Equation (1), the NOx removal rates corresponding to the three temperatures are 9.1%, 10%, and 12%. When the inlet gas is $NO/NO_2/NH_3/N_2$, simulating the fast SCR reaction Equation (2), the NOx removal efficiencies corresponding to the three temperatures are 53%, 69.2%, and 70.8%, respectively. When the inlet gas is $NO/NH_3/O_2/N_2$, simulating the standard SCR reaction Equation (3), the NOx removal efficiencies corresponding to the three temperatures are 34.6%, 16%, and 31.6%, respectively.

(4) There are two main methods for the application of activated carbon in the exhaust gas denitration of marine diesel engines. When ships use high-sulfur fuel oil, the exhaust gas cooled after the wet scrubber desulfurization can be passed into the activated carbon to oxidize NO to $NO_2$, which is more soluble in water. Then the gas can be passed to a wet scrubber for denitration. When ships use low-sulfur fuel oil, the high-temperature exhaust gas can be passed into the activated carbon catalytic reactor. Then NH3 can be injected into the reactor for SCR denitration treatment.

**Author Contributions:** Conceptualization, Z.W., J.Z. and Y.J.; data curation, H.K. and L.C.; formal analysis, Z.W., H.K.; investigation, Z.W., H.K. and L.C.; resources, Y.J.; writing—original draft preparation, Z.W.; writing—review and editing, J.Z. and Y.J.

**Funding:** This research was supported by the National Natural Science Foundation of China (51876019, 51579026), Innovative talent support program for University of Liaoning province (LR2017048), and the Transportation Industry High-Level Talent Training Program.

**Conflicts of Interest:** The authors declare no conflict of interest.

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
