# Peer review of "Nitrogen Oxide Removal by Coal-Based Activated Carbon for a Marine Diesel Engine"

_applsci, doi:10.3390/app9081656_

Round 1
Reviewer 1 Report
The paper reports a study on the NOx removal from exhaust gases, emitted by marine diesel engine using coal-based activated carbon. This manuscript deserves to be published after considering the following points:
- The terms "catalytic" or "catalyst", when applied to AC in this study, should be eliminated since AC is consumed, so it is only a reagent, during the reaction: see Line 17, 20, 21, 85, 86, 147, 149, 155, 159, 188, …..
- The second point concerns the stability. Since the material using for denitration is consumed during the reaction, It is difficult for me to conceive that one can speak of "stability" of the material towards the reaction or of "stable efficiency": see abstract line 17, and line 358. Indeed, with a weak reaction time combined with weak VVH, it is obvious that we will have an impression of “stability”.
- The value of the VVH (volume of gas per volume of catalyst end per hour) used for the experiments has to be report in the experimental part and the gas analysis method has to be described.
- Line 290: 200°C instead of 2000°C and 300°C instead of 3000°C
Reply to the Reviewer's comments
1. The terms "catalytic" or "catalyst", when applied to AC in this study, should be eliminated since AC is consumed, so it is only a reagent, during the reaction: see Line 17, 20, 21, 85, 86, 147, 149, 155, 159, 188, …..
Thanks for the reviewer’s comments.
Firstly, as can be seen from Figure 2, Figure 3 and Figure 5, no CO is generated when the temperature is lower than 100 ℃. And then the CO concentration increases when the temperature is above 200 ℃, especially during the temperature rise from 100 ℃ to 200 ℃ and from 200 ℃ to 300 ℃. However, when the temperature is stable, the CO concentration gradually decreases until it disappears. Thereby it indicates that the activated carbon is not as reactant in the reaction, but mainly as catalyst. The CO produced during temperature changes may be mainly caused by the decomposition of related oxygen-containing functional groups, which is explained in detail in Section 3.2 of the paper.
Secondly, we weighed the activated carbon by a precise balance before and after each experiment. It was found that the quantity of the activated carbon has no noticeable change before and after the experiment. This also means that the activated carbon is not as reactant in the reaction, but mainly as catalyst.
Finally, the literatures [16-18], [23-28] and other studies all believe that activated carbon mainly plays the role of adsorbent and catalyst below 400 ℃ in the desulfurization and denitration process, which as reactant in the denitration process is basically negligible. Activated carbon will be the reducing agent in the denitration process only when the temperature is higher than 700 ℃
In summary, we believe that it is reasonable to discuss the catalytic activity of activated carbon below 300 ℃, and its participation as reactant in the denitration process can be neglected.
2. The second point concerns the stability. Since the material using for denitration is consumed during the reaction, It is difficult for me to conceive that one can speak of "stability" of the material towards the reaction or of "stable efficiency": see abstract line 17, and line 358. Indeed, with a weak reaction time combined with weak VVH, it is obvious that we will have an impression of “stability”.
Indeed, “stable” or “stability” is a strong word. In the AC denitration case, AC plays a combined role of absorbent, reagent and catalyst. It is different at different temperatures. As explained in Question 1, activated carbon as catalyst from room temperature to 300℃ to participate in the denitration process can be determined in this paper. The stable denitration efficiency mentioned in this paper is mainly because of activated carbon is a good adsorbent material, and the adsorption process is gradually saturated with the increase of time.
In order to distinguish from the influence of the adsorption of activated carbon on the denitration efficiency, the term of “stable denitration efficiency” was used. When the denitration efficiency is close to being stable, it means the activated carbon has been saturated adsorption. The denitration efficiency at this time is mainly a result of the catalytic denitration process of activated carbon as the catalyst. However, to be more precise, we added a line to explain this and also changed the term to “near-stable denitration efficiency”. The revised results can be found in Line 17, 128, 129, 157, 202, 263, 324, 329, 348.
3. The value of the VVH (volume of gas per volume of catalyst and per hour) used for the experiments has to be report in the experimental part and the gas analysis method has to be described.
Since the VVH was fixed in this paper, we didn’t discuss the effect of VVH on the denitration efficiency. Therefore, this parameter was not given in this paper. According to the reviewer's suggestion, we added this parameter in line 117.
The corresponding volume of gas per volume of catalyst and per hour, i.e. VVH, was 2500 h-1.
The flue gas analyzer used in this paper was Testo 350, which analyzes the concentration of gas by electrochemical method. According to the reviewer's suggestion, we added the relevant description in lines 102-104.
The flue gas analyzer can continuously monitor the temperature and the concentration of NO, NO2 and CO of the mixed gas with the sampling period of 1 second by electrochemical method.
4. Line 290: 200°C instead of 2000°C and 300°C instead of 3000°C
Thanks to the reviewer. The mistake has been revised as suggested.

Reviewer 2 Report
Very interesting study. The paper is well written and easy to follow. The experimental results clearly show the ability of AC systems to be used as catalysts for diesel exhaust denitration process. The AC catalysts are cheaper and can catalyze the NO to NO2 conversion at lower temperatures compare to the vanadium based catalysts. The experimental results presented in this manuscript are very important and contribute towards optimizing the reaction conditions of denitration catalysts. I highly recommend this article for publication.
There are two typo which should be corrected.
1: Figure 1 caption typo “denigration”
2: Figure 9 caption 200 and 300 C are mistyped as 2000 C and 3000C
Author Response
1. Figure 1 caption typo “denigration”
Thanks to the referee. The word of “denigration” in Figure 1 caption has been replaced by “denitration” as suggested.
2. Figure 9 caption 200 and 300 C are mistyped as 2000 C and 3000C
Thanks to the referee. The mistake has been revised as suggested.